# Numerical Investigation of the Effect of Longitudinal Fiberglass Dowels on Tunnel Face Support in Layered Soils

**Morteza Esmaeili** [1,*] **, Jafar Hosseini Manoujan** [1] **, Jafar Chalabii** [2] **, Farshad Astaraki** [3]
**and Majid Movahedi Rad** [2,*]

1   School of Railway Engineering, Iran University of Science and Technology, Tehran 16846, Iran;
    jafar_hoseini@alumni.iust.ac.ir
2   Department of Structural and Geotechnical Engineering, Széchenyi István University, Egyetem tér 1,
    9026 Győr, Hungary; chalabii.jafar@sze.hu
3   Department of Civil and Environmental Engineering, University of Alberta, Edmonton, AB T6G 2E1, Canada;
    astaraki@ualberta.ca
*   Correspondence: m_esmaeili@iust.ac.ir (M.E.); majidmr@sze.hu (M.M.R.)

**Abstract:** Tunnel face extrusion rigidity is an important factor for solving stress–strain problems in loose ground conditions. In previous studies, the effect of horizontal and vertical soil layering on tunnel excavation face stability in the presence of longitudinal fiberglass dowels has not been studied. Therefore, in this study, the effect of fiberglass dowels on the stability of the tunnel face in layered soil has been investigated. In this matter, the best dowel arrangement for minimizing the excavation face extrusion in the case of two-layer soil (horizontal or vertical) has been focused on. For this purpose, firstly, a 3D numerical model was validated based on field data provided previously, and then a 3D numerical tunnel was developed in FLAC3D, adopting the Mohr–Coulomb failure criterion. In continuation, the effect of tunnel diameter, initial pressure ranging from 0.5 to 1.5 MPa, and different placement angles of fiberglass dowels ranging from 0 to 9 degrees, with respect to the tunnel longitudinal axes on the tunnel face extrusion, have been investigated. In the case of horizontal layering, the results showed that the maximum extrusion rate is significantly increased where the elasticity modulus of the soil is reduced. In addition, comparing the maximum extrusion in vertical and horizontal layering, it was found that its value in the horizontal mode is much higher than in the vertical. Additionally, the extrusion of the tunnel face has changed significantly due to an alteration in the initial stress. Finally, it was discovered that tunnel face extrusion is not significantly affected by altering the angle of the fiberglass dowels.

**Keywords:** face extrusion; fiberglass dowels; excavation face; FLAC 3D

## 1. Introduction

In the context of underground activities, stabilizing the tunnel excavation face with appropriate reinforcement via adequate and efficient doweling is a crucial issue that, if ignored, might result in the face collapsing and consequent financial and property losses. Fiberglass longitudinal dowels increase the stability forces and decrease the instability forces on the excavation face. Therefore, fiberglass tensile dowels are an effective method for improving the excavation face in difficult drilling conditions.

The pre-consolidation measures of the tunnel are applied inside the rock mass and on the excavation face in conditions where triaxial stress conditions still exist. These actions aim to keep the minimal limiting stress from dropping to zero, maintaining the capacity to control displacements. Therefore, it is essential to apply enough restricting pressure ahead of the work front to prevent it before loosening or ruining the land, dealing with challenging circumstances, and engaging in expensive stabilizing measures [1]. Lunardi et al. [2] carried out experimental studies to investigate the effect of fiberglass dowels on the tunnel excavation face. Their main aim was to build a scaled-down physical model

to monitor the behavior of the excavation face in different stress–strain conditions. So, they compared the tunnel faces with and without fiberglass dowels and demonstrated the effect of fiberglass elements on tunnel face behavior. In the same study, conducted by Paternesi et al. [3], the stability of tunnel excavation faces in the presence and absence of fiberglass dowel reinforcement was examined. Their calculations concerning unreinforced faces showed that the basic formulation of the LEM, adopting the Horn mechanism, may significantly overestimate safety factors.

Kamata & Mashimo [4], using a series of lab tests, proved that the displacement of the tunnel excavation face was minimized where the fiberglass dowels were placed perimetrically. Peila [5] found out that placing fiberglass dowels in the excavation faces of the tunnels decreased the extrusion significantly. In a parametric study, accomplished by Ng & Lee [6], the use of fiberglass dowels in the tunnel heading was found to be effective. The results indicated that using the dowels minimized the plastic zone, excess pore water pressure, and surface settlement.

Oreste [7] worked on a computational method for investigating the stability of the excavation face of deep tunnels using longitudinal fiberglass rods. It was found that increasing the fiberglass dowel's length during excavation resulted in decreasing displacement and extrusion. To design tunnel face reinforcement using horizontal doweling, a convergence-confinement approach was suggested by Dias [8]. Also, Perazzelli & Anagnostou [9] presented a computational process based on the limit equilibrium method, which aims to evaluate the stability of the tunnel excavation in cohesion–friction soils.

Kovári & Leonardi [10] investigated a study in which the observational method in tunneling was assessed and examined. This method was conducted through a field survey in which a tunnel excavation face was reinforced by about 70 fiberglass dowel elements on average, with a length and overlap of 15 m and 5 m, respectively. The bolt lengths were determined based on the geotechnical and hydrogeological parameters of the tunnel, as well as by testing different lengths until the optimal ones were found. The performance of fiberglass dowels in the confinement of the tunnel face in weak rock masses was theoretically simulated by Zaheri et al. in 2023 [11]. The distribution of the axial forces and shear stresses along the reinforcements was determined by taking into account the relative shear displacements between the reinforcing elements and the rock mass. Additionally, some examples were resolved, and the outcomes were compared with those obtained from FLAC3D simulations. The findings obtained using both methods outlined were found to be in good agreement.

In 2020, Han et al. [12] suggested an analytical prediction model after applying the limit analysis approach to examine the stability of the tunnel face. They also looked at the beneficial impacts of steel pipe umbrellas and longitudinal fiberglass dowels on the stability of the tunnel face. Additionally, a sensitivity analysis was carried out to ascertain how the tunnel design, the space between installing reinforcement, the depth of the cover, and the reduction factor influenced the required limit reinforcement density. In 2022, Wang et al. [13] developed a closed-form solution based on the theory of spherical symmetry inhomogeneous initial stress for the study of face extrusion deformation with reinforcement during full-face tunnel excavation. By comparing the results of the suggested model with those of the numerical simulation, they verified its accuracy and applicability. The results of the aforementioned investigation showed that when different parameters were increased, the extrusion deformation steadily decreased. This indicated that increasing the face's strength significantly contributed to ensuring its stability.

Furthermore, Zhang et al. [14] analyzed the Yenikapi-Unkapani tunnel's stability in 2020 and established the viability of the suggested model. In order to explore the effects of face bolts and an umbrella arch on the stability of a tunnel face and an unsupported span, they finally conducted parametric studies. The influence of bolt layouts, including the length and density of the bolts, and seepage conditions, including the groundwater level and permeability anisotropy, was investigated using parametric analyses by Hou et al.

in 2023. For reference, they also offered a number of design nomograms for varied soil strength and water level characteristics [15].

The stability of full-face tunnels with weak surrounding rock was analyzed in 2020 by Chen et al. [16]. They also established a relationship between plastic failure depths and buried depths, looked into how the plastic failure zone expanded with increasing buried depths, and proposed a workable method for figuring out the overlap length of the glass fiber reinforcement anchor for tunnel face stability. Also, in 2022, Sun et al. [17] proposed a practical method to simulate the soil strain softening effect in tunnel face stability analysis. They extended the elastoplastic Mohr–Coulomb constitutive model, compared the results with existing experiments, and summarized a collapse mechanism for tunnel faces in sandy soil based on the observed failure-zone growth and decreased shear strength in the softened zone.

In order to determine the limit support pressure and take into account both the vertical and horizontal effects of soil arching, Ye et al. proposed a slicing technique based on the wedge-prism model in 2022 [18]. They explored the role of the incomplete soil arching effect and introduced the rotation of the primary stress. It was validated and demonstrated that the modified model agrees with theoretical and numerical models. To fully comprehend the suggested model, the study additionally examined factors related to the limit support pressure. In addition, in 2023, Lu et al. [19] introduced a protective approach to safeguard tunnel structures. So, static and dynamic experiments were conducted to assess the unconfined compressive strength (UCS), flexural, and compaction resistance at various mixing ratios. The results indicate that the addition of porous sand decreases the UCS compared to the solid sand under similar mixing conditions.

The effects of fiberglass bolts when the tunnel face comes into contact with two different soil layers—vertical or horizontal layers—have not been studied, according to previous research. The study uses a validated and developed FE model based on Dias's research to compare the non-reinforced and reinforced tunnels [8]. The study also analyzes how the tunnel diameter, initial pressure, and angle of the fiberglass bolts influence the maximum displacement of the tunnel face in the non-doweled and doweled conditions. In the drilling excavation, two soil layers with varying mechanical qualities are taken into account.

## 2. Validation and Development of the 3D Numerical Model

In the Firenzuola tunnel in Italy, where studies were conducted by Leonardi in 2008 [1], fiberglass bolts were used for the stability of the tunnel face based on the pattern shown in Figure 1.

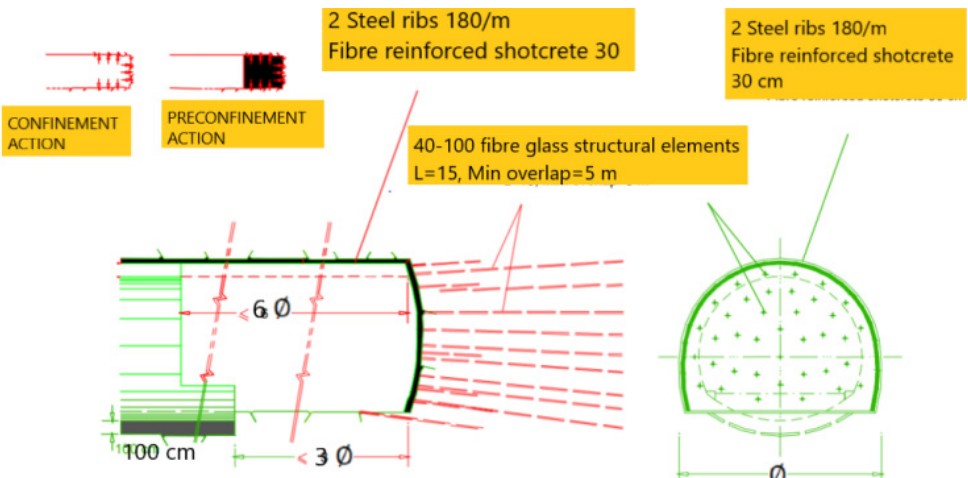

**Figure 1.** Firenzuola Tunnel in Italy.

The final goal of employing fiberglass longitudinal elements to reinforce the core excavation in tunnels is to improve the strength of the soil mass and avoid excessive extrusion of the tunnel excavation, which results in the stability of the drilling excavation even in weak rock masses. This method includes the horizontal placement of fiberglass dowels in the drilling excavation face. The dowels are connected continuously to the ground and the surrounding rocks inside the holes by injecting mortar. They also undergo tension, shear, and bending loading. The reinforcing elements mentioned are then cut during the drilling process. In recent years, even in extremely challenging geotechnical circumstances, this reinforcing technique has often been employed, especially in Italy and France [1]. The fiberglass elements are usually hollow rebars and to prevent tubular reinforcement from slipping out, its surface is usually cut with a spiral groove that increases its friction, and this is made of heat-resistant polyester resin reinforced with glass fibers [1]. This technology can be used in cohesive and relatively cohesive soils, and it can even be used In soils with poor cohesion if the safety of drilling holes is considered [1]. A significant improvement has been reported in the stress–strain characteristics of core-tunnel excavation by implementing this method. A doweling design should be defined based on important parameters such as quantities, length, and the cross-section of elements required for the stability of drilling excavation with a certain safety margin. The technology discussed, "dry drilling," is the number of holes that are drilled relatively parallel to the axe of tunnel uniform distribution in the excavation face, and are generally bigger than the diameter of the tunnel [1]. The following series of reinforcing operations will be carried out if, after the progress of the tunnel, the remaining length of the reinforced area is unable to provide the necessary pre-limiting pressure to continue drilling (the condition that can be immediately identified after reading the amount of extrusion) [1]. The surfaces of tubular fiberglass have spiral grooves to increase their slip resistance. Tubular fiberglass has a limit length of about 15 to 18 m. Fiberglass rods are taller, and they reach ten meters in length. If these resistors are connected, it is necessary to use circular connectors and glue (epoxy resin) to provide tension and shear strength. Fiberglass dowels with different lengths and positions are used depending on the condition of the excavation face and the cross-sectional area of the tunnel [1]. By using these dowels, in addition to reducing the displacement of the excavation face, the magnitude of the crown and wall displacement is also reduced [20]. The optimal length of doweling is very important, so if the length of the dowels is more than the optimal length, it will result in an uneconomical design, and if the length is less than the optimal, it will increase the deformation of the excavation face and even the instability of the tunnel [21].

In the current research, the simulation has been performed by FLAC3D V5, a finite difference software, and a collection of Itasca that is used for continuous environments. The software is based on the Lagrangian computational analysis, which is also suitable for modeling large deformations. The behavior of soil, stone, or other materials that have plastic behavior can be simulated by the software when reaching the yield point. The method mentioned has been validated by the results of Dias's research [8]. In Dias's research, the tunnel is circular with a diameter of 11.6 m.

Due to symmetry, only a quarter of the total cross-section of the tunnel is considered for analysis. The initial state of the stresses is assumed to be homogeneous and isotropic. The mechanical properties of the parameters are presented in Table 1. The entire model includes approximately 20,000 zones and 800 structural elements. The mesh width in the X and Z directions was six times the diameter (approximately 80 m) and in the Y direction was seven times the diameter (approximately 90 m). Figure 2 presents the validation results, which indicate that the validation results of the authors are in good agreement with the results of Dias's simulation [8]. The modeling error also depends on various parameters, such as boundary conditions, model geometry, geomechanical parameters, etc.

**Table 1.** Parameters used for three-dimensional analysis.

| Parameter | Unit | Rock Mass | Maintenance System | Fiberglass Dowel |
|---|---|---|---|---|
| Young's modulus | Mpa | 300 | 10,000 | 20,000 |
| Poisson's ratio | — | 0.3 | 0.2 | — |
| Friction angle | Degree | 20 | — | — |
| Cohesion | Kpa | 50 | — | — |
| Thickness | m | — | 0.2 | — |
| Initial stresses | Mpa | 0.8 | — | — |
| Cross section | $m^2$ | — | — | 0.0014 |
| Pull Limit | t | — | — | 70 |

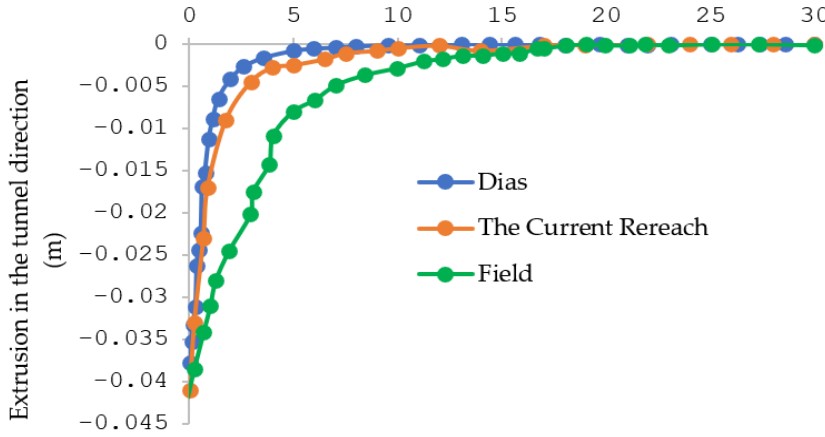

**Figure 2.** The results of validation.

## 3. Layered Soil Simulation

In this section, using the validated model, a parametric study has been performed to investigate the behavior of the excavation face of circular tunnels in the case of two-layer soil (weak soil and collapsing soil) with different elastic moduli in the tunnel excavation face. The tunnelling process while using fiberglass dowels is shown in Figure 3.

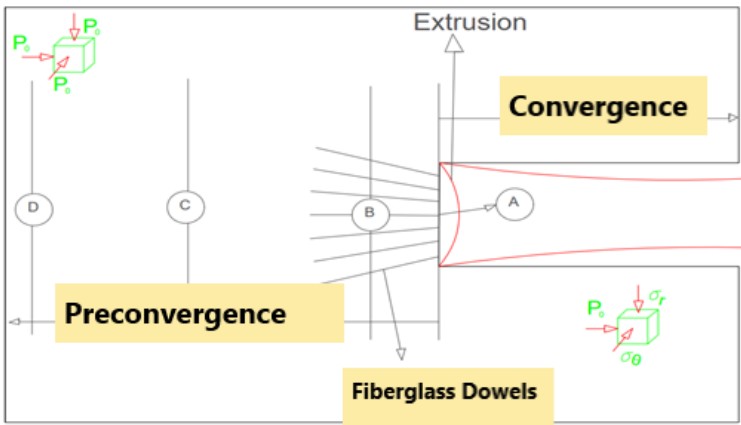

**Figure 3.** Tunnelling process.

In this case, the best dowel arrangement for minimizing the excavation face extrusion in the case of a two-layer soil (horizontal or vertical) is displayed in Figure 4. For this purpose, a 3D numerical model of the tunnel has been developed in FLAC3D, adopting the Mohr–Coulomb failure criterion. In addition, the effects of tunnel diameter, initial pressure,

and different placement angles of fiberglass dowels, with respect to the tunnel longitudinal axes on the tunnel face extrusion, have been investigated.

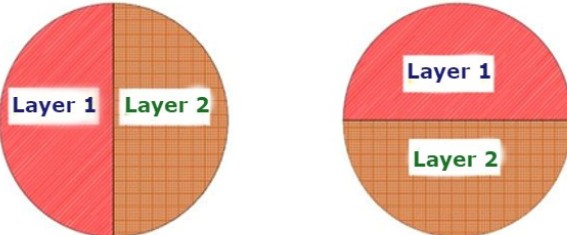

**Figure 4.** Excavation face extrusion in the case of a two-layer soil.

Different modes are considered for the layering and elastic modulus of the soil. In each case, the maximum extrusion of the excavation face is calculated. According to the maximum extrusion, the tunnel face is stabilized by fiberglass dowels, which counteract the displacement caused by the excavation. Kirsch's [22] research shows that if the displacement of the drilling excavation in this type of geological material is about one percent of the tunnel diameter, the excavation face of the tunnel is stable. On the other hand, SELI Company [23], in its reports, has proposed an allowable displacement limit of the tunnel in soft fields less than one percent of the tunnel radius. In this study, less than one percent displacement of the tunnel radius will be considered, and the fiberglass dowels required to stabilize the tunnel drilling excavation will be investigated. The tunnel's diameter is 10 m. Also, the Young's modulus, soil cohesion, and friction angle (first and second layers) used for the simulation are 75 to 600 MPa, 50–200 kPa, and 15–25 degrees, respectively (according to geomechanical classification by Bieniawski) [24]. These mechanical properties of the soil are considered to be in the range of moderate to poor mechanical parameters. The parameters used for 3D analysis, specifications of fiberglass dowels, and maintenance system are shown in Table 2. The doweling pattern has been formed radially at the cross-section of the tunnel, as is evidenced in Figure 5. In addition, the doweling densities used are considered to be 0.25, 0.5, 0.75, 1.25, and 1.5 dowels per square meter, respectively, and according to the results of each dowel's density, the number of dowels required for the stability of the tunnel excavation in each case is calculated. It should be mentioned that these amounts represent an attempt to reach the greatest stability of the tunnel face, similar to how they have previously attempted to achieve the best density of fiberglass dowels in earlier works.

**Table 2.** Parameters used for three-dimensional analysis.

| Parameter | Unit | Fiberglass Dowel | Soil | Maintenance System |
|---|---|---|---|---|
| Diameter | m | 0.042 | — | — |
| Length | m | 15 | — | — |
| Young's modulus | Mpa | 20,000 | 75–600 | 10,000 |
| Cohesion | Kpa | — | 50–200 | — |
| Friction angle | Degree | — | 15–25 | — |
| Tensile strength | KN | 500 | — | — |
| Slurry hardness | Mpa | 10,000 | — | — |
| Hole diameter | m | 0.052 | — | — |
| Poisson's ratio | — | — | 0.15–0.5 | 0.2 |

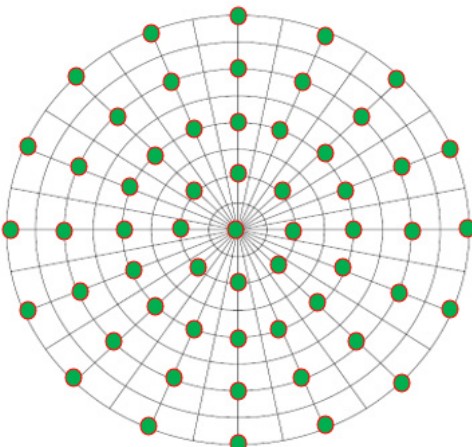

**Figure 5.** Doweling pattern.

The values of initial isotropic and homogeneous stress have been considered, and they are 1 MPa; this value is the average stress state encountered in the Tehran metro tunnel [23]. The model is shown in Figure 6. Due to the symmetry of the tunnel, half of it has been modeled. The built model's dimensions are $80 \times 70 \times 70$ m in length, height, and width, respectively. It should be clarified that these dimensions are selected based on sensitivity analyses that are 5 to 10 times the tunnel diameter to prevent the effects of boundaries.

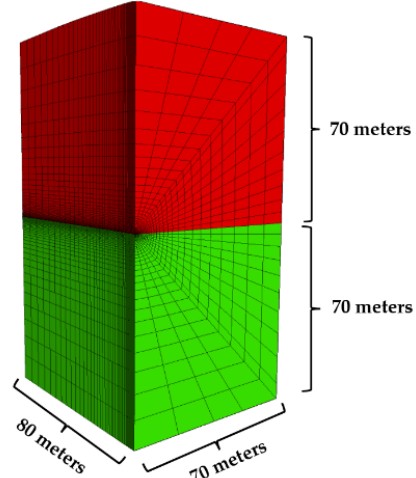

**Figure 6.** Built model for horizontal and vertical layering.

The physical limitations and main conditions have been explained using boundary and initial conditions. In this research, the model's dimensions and the floor of the model have been fixed in the X and Y directions, and only the displacement in the vertical direction has been allowed, as shown in Figure 7. According to the purpose of modeling, deformations, and desired stresses, as well as the investigated area, the number of elements is different. Finally, according to the extracted results, the number of elements was approximately 32,000 zones.

The soil and fiberglass dowels have been modeled elasto-plastically according to the Mohr–Coulomb criterion and there is full contact between the dowels and the soil around them, and the maximum friction at the interface depends on the soil's internal friction angle. The analyses assume that there is no slippage between the nail and the soil. It should be noted that the fiberglass dowels have been modeled by cable structural elements. In addition, the concrete holder used was elastic. The fiberglass dowels are 15 m in length and are divided into 15 parts. It should be noted that this length and the divided part are based on previous works carried out by Aksoy [20].

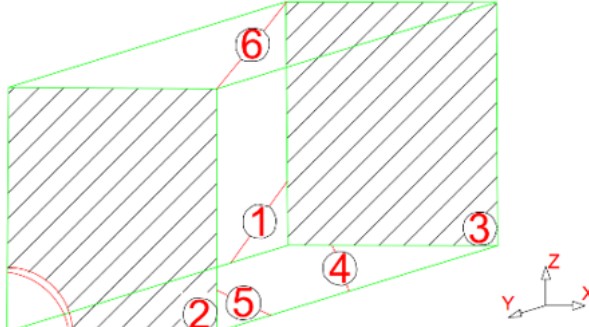

**Figure 7.** Boundary conditions.

Furthermore, the length of their overlap is 6 m [23]. Drilling will be performed in three stages; in each stage, the length of drilling is 2 m. Moreover, two meters of fiberglass dowels will be cut in each stage. The fiberglass dowels have been provided for each case according to the maximum extrusion. The tunnel has been maintained at the same time as the drilling process. Also, drilling has been performed in a full cross-section by the rotary drum cutters. After the last drilling stage, the maximum amount of extrusion, vertical displacement of the tunnel crown, settlement, and longitudinal displacement are measured. The effect of different parameters on the maximum extrusion of the tunnel excavation face will be investigated in the following sections. In addition, the selection of the fiberglass dowel quantity required for its stability will be studied. In this case, the elasticity modulus of the first layer is equal to 300 MPa [23]. Furthermore, the elasticity modulus of the second layer in three stages of simulation has been considered to be 1/2, 1/3, and 1/4 of the first layer's elasticity modulus, respectively, to study the changes in the tunnel excavation's extrusion due to changes in the elasticity modulus of the second layer while keeping the first layer's elasticity modulus constant.

## 4. Effect of Elasticity Modulus in Horizontal Layering

### 4.1. First Model

In this model, two layers of soil with different elasticity moduli are located in the excavation face of the tunnel. It is necessary to mention that the soil layer with a higher elasticity modulus (the first layer) is located above the soil layer with a lower elasticity modulus (the second layer). In addition, by reducing the elasticity modulus of the second layer and keeping the elasticity modulus of the first layer constant (Figure 4), the behavior of the layers located in the tunnel excavation face will be investigated. Figure 8 shows the maximum extrusion in the case without dowels and after placing fiberglass dowels in the tunnel's excavation face. As can be seen, different numbers of fiberglass elements were applied to the tunnel face, and so the amount of extrusion in the excavation face has been significantly reduced by the placement of fiberglass dowels.

It should be mentioned that, as shown in Figure 8, by decreasing the second layer's elasticity, the maximum extrusion value of the tunnel's excavation face increases, in which case more dowels will be needed to stabilize the tunnel face. Moreover, since the second layer's elasticity modulus is lower than the first layer's, its displacement and maximum extrusion will be greater. As a result, this layer will need more dowels for stability.

Table 3 presents the values of the maximum tunnel's extrusion in the case without reinforcement and the use of fiberglass dowels. Moreover, the number of fiberglass dowels required to stabilize the tunnel excavation has been presented. According to this table, by decreasing the elasticity modulus, the amount of extrusion has increased, and therefore more dowels will be needed to stabilize the tunnel excavation.

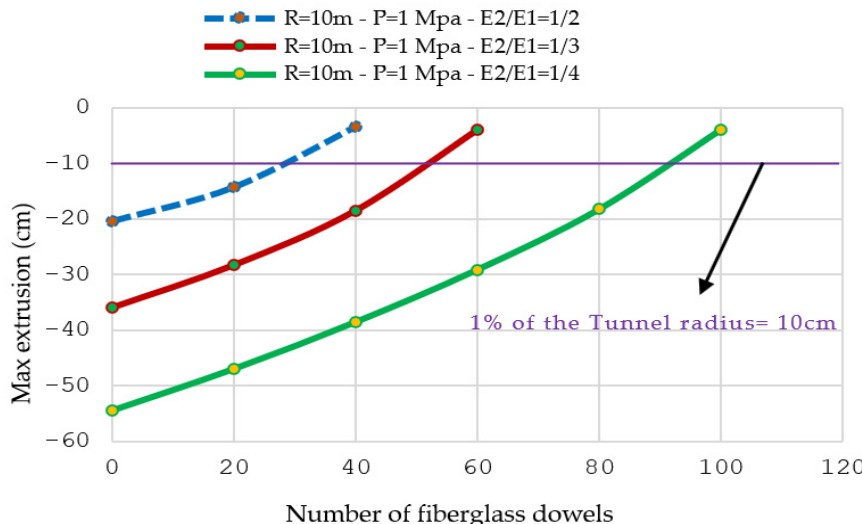

**Figure 8.** Maximum extrusion of the tunnel excavation in non-doweled and doweled model (model 1).

**Table 3.** Maximum extrusion and number of fiberglass dowels to stabilize the tunnel excavation (model 1).

| Elasticity Modulus Ratio | ME [1] without FE [2] (cm) | ME with FE (cm) | NFEFL [3] | NFESL [4] |
|---|---|---|---|---|
| E2/E1 = 1/2 | −20.035 | −3.35 | 14 | 26 |
| E2/E1 = 1/3 | −36.0 | −4.02 | 20 | 40 |
| E2/E1 = 1/4 | −54.5 | −3.89 | 28 | 72 |

[1] Maximum Extrusion; [2] Fiberglass Elements; [3] Number of Fiberglass Elements in the First Layer; [4] Number of Fiberglass Elements in the Second Layer.

### 4.2. Second Model

In this case, unlike the first case, the soil layer with a lower elasticity modulus (the first layer) is placed on top of the soil layer with a higher elasticity modulus (the second layer). So, by reducing the elasticity modulus of the first layer and keeping the elasticity modulus of the second layer constant, the behavior of the layers located in the tunnel excavation face will be investigated. According to the results, the maximum extrusion of the tunnel excavation has significantly decreased in contrast to the first model, and the effect of layering is visible. Figure 9 shows the maximum extrusion in the case of non-reinforced and reinforced models.

Furthermore, as shown in this case, fewer dowels will be needed to stabilize the tunnel excavation face. Also, since the first layer's elasticity modulus is lower than the second layer's, its displacement and maximum extrusion will be greater, and as a result, this layer will need more dowels for stability.

Table 4 presents the values of the maximum tunnel excavation's extrusion in the case without reinforcement and the use of fiberglass dowels, as well as the number of them required to stabilize the tunnel excavation. It is obvious that by decreasing the elasticity modulus, the extrusion of the tunnel face is increased. It can be seen that, in comparison to case 1, the maximum tunnel excavation face extrusion is significantly decreased. According to this table, the amount of extrusion has been increased by decreasing the modulus of elasticity, and more dowels will be needed to stabilize the tunnel excavation.

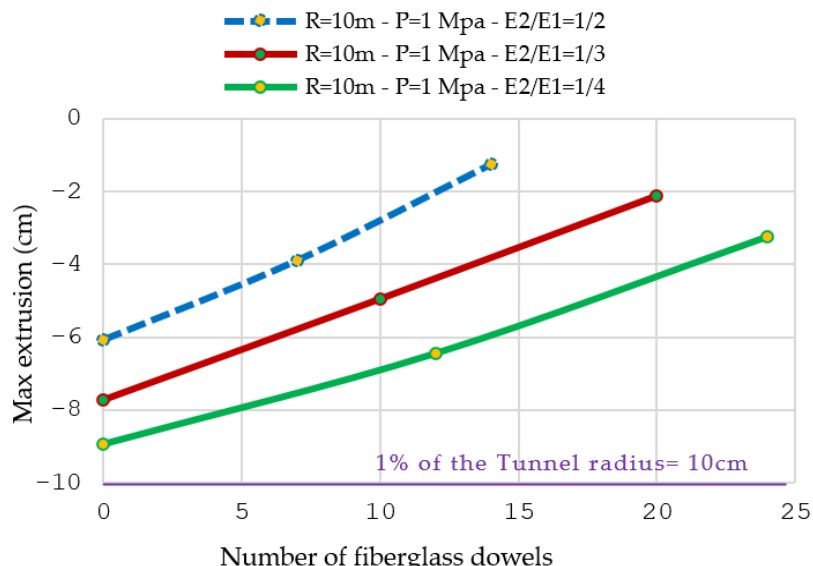

**Figure 9.** Maximum extrusion of the tunnel excavation in non-doweled and doweled model (model 2).

**Table 4.** Maximum extrusion and number of fiberglass dowels to stabilize the tunnel excavation (model 2).

| Elasticity Modulus Ratio | ME without FE (cm) | ME with FE (cm) | NFEFL | NFESL |
|---|---|---|---|---|
| E2/E1 = 1/2 | −6.08 | −1.24 | 4 | 10 |
| E2/E1 = 1/3 | −7.73 | −2.11 | 10 | 10 |
| E2/E1 = 1/4 | −8.95 | −3.23 | 16 | 8 |

## 5. Effect of Elasticity Modulus in Vertical Layering

In this case, two soil layers have been placed vertically within the tunnel excavation face. Changing the behavior of soil layers located in the tunnel face has been investigated by reducing the second layer's elasticity modulus and keeping the first layer's elasticity modulus constant. In this respect, according to the results of the numerical simulation, the maximum extrusion of the tunnel face is less than 1% of the diameter, but to investigate the effect of fiberglass dowels on reducing extrusion, fiberglass dowels have been placed in the tunnel excavation face.

Figure 10 illustrates the maximum extrusion in the case of non-dowelled and dowelled tunnel excavation faces. As it is known, the amount of excavation face extrusion has been reduced by the placement of fiberglass dowels. In addition, as shown in Figure 10, the maximum tunnel face extrusion has been increased by decreasing the second layer's elasticity modulus. Moreover, the second layer's displacement and maximum extrusion will be greater because the second layer has a lower modulus of elasticity than the first layer.

Table 5 provides the amount of the maximum tunnel face's extrusion for the non-reinforced model and the model reinforced with fiberglass dowels, as well as the number of dowels required to stabilize the tunnel excavation.

**Table 5.** Maximum extrusion and number of fiberglass dowels to stabilize the tunnel excavation (model 3).

| Elasticity Modulus Ratio | ME without FE (cm) | ME with FE (cm) | NFEFL | NFESL |
|---|---|---|---|---|
| E2/E1 = 1/2 | −1.83 | −0.8 | 6 | 6 |
| E2/E1 = 1/3 | −3.02 | −1.01 | 6 | 10 |
| E2/E1 = 1/4 | −4.58 | −1.45 | 8 | 12 |

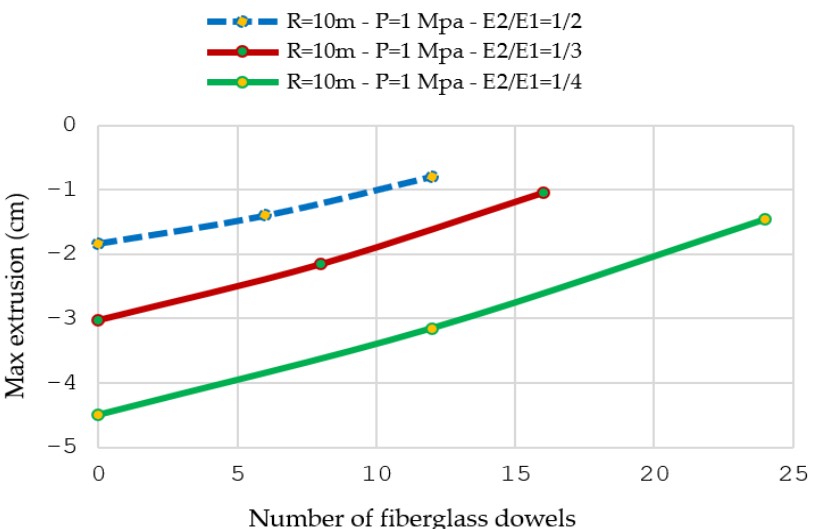

**Figure 10.** Maximum extrusion of the tunnel excavation in non-reinforced and reinforced models (vertical layering).

## 6. Variations in Initial Pressure

Three initial pressures of 0.5, 1, and 1.5 MPa have been considered to investigate the effect of the initial pressure change on the maximum extrusion of the tunnel face. Figure 11 provides the maximum extrusion of the tunnel excavation face for the three initial pressures mentioned. It also shows the cases of non-reinforced and reinforced models. According to this figure, it can be recognized that the extrusion of the tunnel face has been changed substantially by changing the initial pressure value. On the other hand, the amount of tunnel excavation face extrusion has increased significantly by increasing the initial pressure.

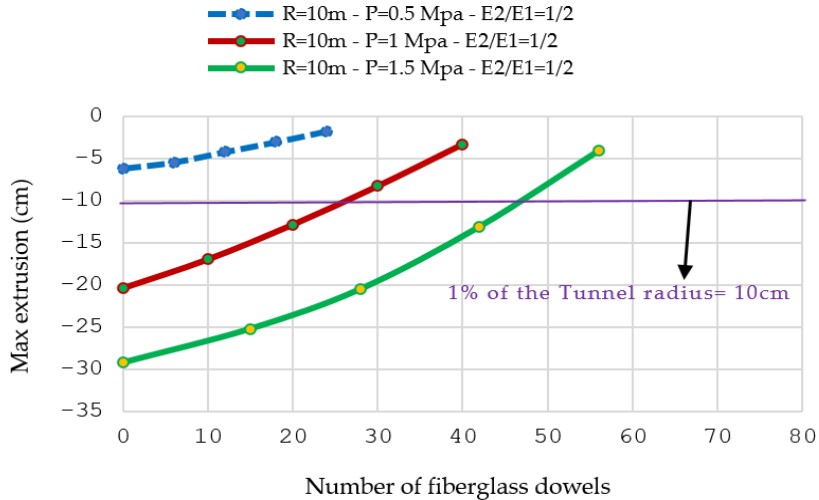

**Figure 11.** Maximum extrusion of the tunnel excavation due to the change in initial pressure in the non-doweled and doweled models (two different layers in the tunnel excavation face).

## 7. The Effect of Fiberglass Dowels' Placement Angle on the Maximum Extrusion of the Tunnel Excavation

Four different angles of 0, 3, 6, and 9 degrees have been considered to investigate the effect of changing the fiberglass dowels' placement angle on the maximum extrusion of the tunnel excavation face. Figure 12 illustrates the maximum tunnel face extrusion in terms of changing the fiberglass dowel placement angle from 0 to 9 degrees. It also shows the maximum tunnel extrusion in the cases of non-doweled and doweled models. As shown in this figure, the tunnel excavation extrusion has changed insignificantly by changing the

fiberglass dowel placement angle, and these changes have been in the range of 0 to 1 cm. Accordingly, it can be concluded that changing the fiberglass dowel's placement angle has a minor effect on the maximum tunnel extrusion, so this parameter can be omitted when placing fiberglass dowels within the tunnel excavation face.

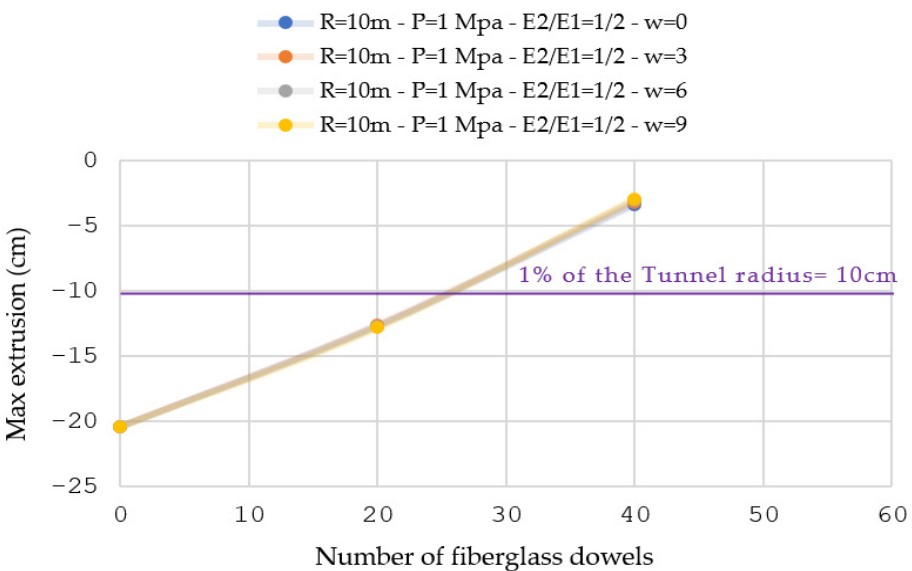

**Figure 12.** Maximum tunnel excavation extrusion due to the change in dowels' angles in the non-doweled and doweled models.

## 8. Comparison of Maximum Extrusion of Horizontal and Vertical Layering

As observed, in the case that the tunnel excavation layering is horizontal, if the soil with a higher modulus is located on the weaker part, the changing rate of extrusion is higher than when the layer with a high elasticity modulus is below the weaker part. When soil with a higher modulus is placed on soil with a lower modulus, due to the low shear strength of the bottom layer and the amount of pressure applied by the top layer, the ductility and its shape change increase. Furthermore, in the second case, when the softer layer is placed on top, the extrusion is significantly reduced.

In the case where the two layers are placed vertically within the tunnel excavation face, the layers' pressure on each other will no longer be discussed, and, as a rule, the deformation will be less than in the horizontal mode. Also, in this situation, the layer that has a lower modulus will have more extrusion. It should be mentioned that by observing and investigating the maximum extrusion of horizontal and vertical layering in the tunnel excavation face, the result obtained shows that in the case where the tunnel excavation face is horizontal and the layers are stacked on top of each other, the maximum extrusion of the excavation face will be significantly higher than that in the vertical layering model, which is shown in Figures 13 and 14.

Tables 6 and 7 show that the amount of extrusion has been increased in both vertical and horizontal layering by decreasing the soil elasticity modulus, which is more obvious in horizontal layering than vertical layering.

**Table 6.** Comparison of maximum extrusion of horizontal layering and vertical layering (Non-Reinforced).

| Elasticity Modulus Ratio | The Maximum Tunnel Face Extrusion (cm) | |
|---|---|---|
| | Vertical Layering | Horizontal Layering |
| E2/E1 = 1/2 | −1.83 | −20.35 |
| E2/E1 = 1/3 | −3.02 | −36 |
| E2/E1 = 1/4 | −4.58 | −54.5 |

**Table 7.** Comparison of maximum extrusion of horizontal layering and vertical layering (Reinforced).

| Elasticity Modulus Ratio | The Maximum Tunnel Face Extrusion (cm) | |
| :---: | :---: | :---: |
| | Vertical Layering | Horizontal Layering |
| $E_2/E_1 = 1/2$ | −0.8 | −3.35 |
| $E_2/E_1 = 1/3$ | −1.01 | −4.02 |
| $E_2/E_1 = 1/4$ | −1.45 | −3.89 |

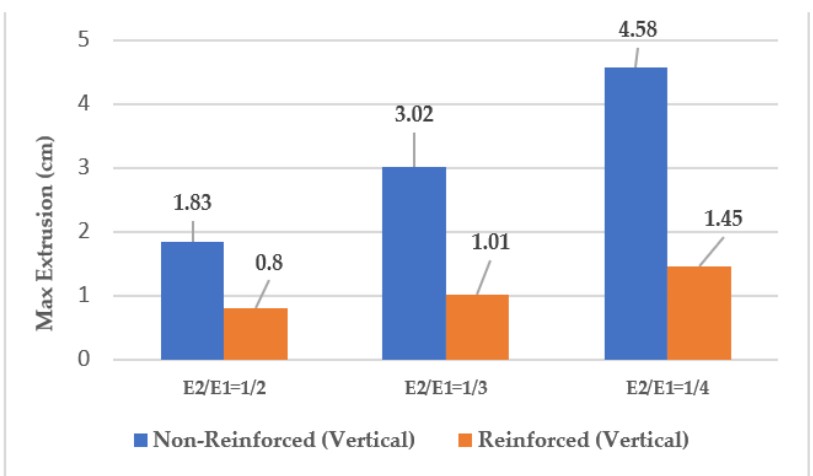

**Figure 13.** Maximum tunnel excavation extrusion due to the change in dowels' angles in the non-doweled and doweled models (Horizontal Layering).

**Figure 14.** Maximum tunnel excavation extrusion due to the change in dowels' angles in the non-doweled and doweled models (Vertical Layering).

## 9. Concluding Remarks

In this study, the influence of fiberglass dowels on tunnel face stability has been numerically investigated in a situation where the properties of the materials in the tunnel excavation face are different (two soil layers with different mechanical properties). Additionally, the effects of initial pressure and angular fiberglass dowels on the maximum extrusion of the tunnel excavation face have been discussed.

When two layers of soil are horizontally located in the tunnel excavation face, when the soil with a higher elasticity modulus is placed on the soil with a lower elasticity modulus, displacements and extrusions are higher than in the soil with a lower elasticity modulus.

As a result, more fiberglass dowels will be needed to stabilize the tunnel excavation, for example, 100 fiberglass dowels with a 54.5 cm extrusion.

If two layers of soil are located vertically in the tunnel excavation face, the impact of the two layers will be less than the horizontal layering, and, as a result, the amount of displacement and tunnel face extrusion will be much less than in the horizontal two-layer model. However, by changing the initial pressure, the tunnel face extrusion will change significantly, so the number of fiberglass dowels required to stabilize the tunnel excavation face will also increase. The maximum extrusion has decreased to 14 cm by decreasing the initial pressure to 0.5 MPa. Moreover, the maximum extrusion has increased to 10 cm by increasing the pressure to 1.5 MPa.

Finally, the amount of extrusion will change slightly when the angle of the fiberglass dowels on the tunnel excavation face is changed.

The study faced some limitations. One of them was obtaining enough data for the validation process, which took several months to achieve. Another challenge was the complexity of the modeling process in FLAC 3D, which required a high level of expertise and skill to simulate the study scenario.

For future research, the following two key suggestions are offered: investigate the effect of different layering patterns on the tunnel face (more than two layers), and find analytical solutions and methods to study how the layering of the tunnel face influences the number of fiberglass bolts needed for each layer.

**Author Contributions:** Writing—review and editing, M.E., M.M.R. and F.A.; conceptualization, M.M.R., J.C. and F.A.; investigation, M.M.R., J.C. and F.A.; supervision, validation, M.E.; software, data curation, methodology, J.H.M.; writing—original draft J.C. All authors have read and agreed to the published version of the manuscript.

**Funding:** This research received no external funding.

**Data Availability Statement:** The datasets are present in the work.

**Conflicts of Interest:** The authors declare that they have no known competing financial interest or personal relationship that could have appeared to influence the work reported in this paper.

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
