# Peer review of "Numerical Investigation of the Effect of Longitudinal Fiberglass Dowels on Tunnel Face Support in Layered Soils"

_infrastructures, doi:10.3390/infrastructures8100138_

Round 1

Reviewer 1 Report

The effect of fiberglass dowels on the stability of the tunnel face in layered soil has been investigated in this study. In this matter, the best dowel arrangement for minimizing the excavation face extrusion in the case of two-layer soil 17 (horizontal or vertical) has been focused. In general, the paper is well structured and written. Some problems can be addressed before acceptance:

1.      The citation and reference are not exactly matched, please double check.

2.      The language of the text should be carefully checked again.

3.      I believe the following paper might be helpful to improve the quality of the paper “Preliminary Mechanical Evaluation of Grouting Concrete as a Protective Layer for Tunnelling”

4.      Table 1, should be MPa, kPa. Same for Table 2, etc.

5.      The conclusion is too long, it needs to be shortened.

6.      Please highlight the key contribution of the study in the abstract.

Reviewer 2 Report

This study investigates the effect of longitudinal fiber-glass dowels on supporting the tunnel face in layered soils. The stability of the tunnel face in layered soil has been investigated, and the best dowel arrangement for minimizing the excavation face extrusion has been determined. The study finds that the presence of horizontal and vertical soil layering affects the stability of the tunnel excavation face in the presence of longitudinal fiberglass dowels. The study provides practical implications for improving the rigidity of the core extrusion in loose ground conditions.

The paper exhibits a commendable structure and composition, while the research itself demonstrates novelty and importance in the field. Prior to publication, I have some minor suggestions to address:

In lines 47-48, it is advisable to refrain from employing abbreviations throughout the academic manuscript.

Consider breaking down the introduction into more concise paragraphs for improved readability and flow.

The final paragraph of the introduction could benefit from rephrasing for greater clarity.

Regarding line 68, it would be valuable to elucidate how the dowel lengths were chosen.

Line 139 could be enhanced by specifying the intended meaning of "straightness."

In Figure 2, refine the formatting by eliminating external borders ensuring font style, color, and size uniformity.

In Figure 4, alter the font color of "Layer 1" for better legibility in print.

Figure 5. Why is the doweling pattern not symmetrical? Since the simulation model assumes symmetry, should this pattern be symmetrical?

Correct the reference errors in lines 248 and 336.

Detail the modeling of the dowel-soil friction mechanism and outline any underlying assumptions in lines 265-270.

To indicate the previously mentioned threshold (1%), consider denoting the corresponding extrusion with a horizontal patterned line in Figures 8-12.

Explain the rationale behind the observation that E2/E1=1/4 necessitates fewer NFESL than the other cases in Table 4. This could involve elaboration on why a lower modulus demands a greater number of fiberglass elements in the second layer compared to the other cases.

In Figures 13 and 14, consider employing a 2D bar chart instead of a 3D bar chart, as the depth does not represent a variable.

Reconsider mentioning the extrusion range of 8cm to 48cm in the initial conclusion, as these values are specific to the assumed geometry and mechanical properties.

Conclude with a paragraph outlining future prospects envisioned by the authors and potential extensions of this work. Additionally, addressing some of the study's limitations would enhance its comprehensiveness.

None

Round 2

Reviewer 1 Report

The paper can be accepted